# A Modification Fractional Homotopy Perturbation Method for Solving Helmholtz and Coupled Helmholtz Equations on Cantor Sets

**Dumitru Baleanu** [1,2] **and Hassan Kamil Jassim** [3,*]

[1] Department of Mathematics, Faculty of Art and Sciences, Çankaya University, Ankara 06530, Turkey; dumitru@cankaya.edu.tr
[2] Institute of Space Sciences, Magurele-Bucharest 077125, Romania
[3] Department of Mathematics, Faculty of Education for Pure Sciences, University of Thi-Qar, Nasiriyah 64001, Iraq
[*] Correspondence: hassan.kamil@yahoo.com

**Abstract:** In this paper, we apply a new technique, namely, the local fractional Laplace homotopy perturbation method (LFLHPM), on Helmholtz and coupled Helmholtz equations to obtain analytical approximate solutions. The iteration procedure is based on local fractional derivative operators (LFDOs). This method is a combination of the local fractional Laplace transform (LFLT) and the homotopy perturbation method (HPM). The method in general is easy to implement and yields good results. Illustrative examples are included to demonstrate the validity and applicability of the new technique.

**Keywords:** Helmholtz equation; local fractional homotopy perturbation method; local fractional Laplace transform; local fractional derivative operator

## 1. Introduction

The theory of local fractional calculus was successfully utilized to describe the non-differentiable problems arising in mathematical physics, such as Schrödinger equations [1], the gas dynamic equation [2], the telegraph equation [3], the wave equation [4–7], Fokker–Planck equations [8,9], Laplace equations [10], Klein–Gordon equations [11], Helmholtz equations [12], and the Goursat problem [13] on Cantor sets.

Several analytical and numerical methods have been used to solve partial differential equations (PDEs) with local fractional derivative operators (LFDOs), such as the Adomian decomposition method [13–15], variational iteration method [16–22], differential transform method [23,24], series expansion method [25], Sumudu transform method [26], Laplace transform method [27], reduced differential transform method [28], Laplace variational iteration method [29], Fourier series method [30], and homotopy perturbation method [31]. Our aim is to present the coupling method of local fractional Laplace transform (LFLT) and homotopy perturbation method (HPM), which we call the local fractional Laplace homotopy perturbation method (LFLHPM), and use it to solve differential Helmholtz and coupled Helmholtz equations on Cantor sets within a local fractional operator.

This paper is organized as follows. In Section 2, the basic mathematical tools of local fractional calculus are introduced. The analysis of the proposed method is given in Section 3. Then, in Sections 4 and 5, the proposed method is implemented to solve differential Helmholtz and coupled Helmholtz equations on Cantor sets within a local fractional operator. Finally, concluding remarks are presented in Section 6.

## 2. Mathematical Fundamentals

**Definition 1.** *The local fractional derivative of $\varphi(\mu)$ of order $\delta$ at the point $\mu_0$ is given by [19,20,32,33]:*

$$\varphi^{(\delta)}(\mu_0) = \lim_{\mu \to \mu_0} \frac{\Gamma(1+\delta)[\varphi(\mu) - \varphi(\mu_0)]}{(\mu - \mu_0)^\delta}, \ 0 < \delta \le 1. \tag{1}$$

**Definition 2.** *Let $\frac{1}{\Gamma(1+\delta)} \int_0^\infty |\varphi(\mu)| (d\mu)^\delta < k < \infty$. The local fractional Laplace transform of $\varphi(\mu)$ is given by [19,20,32,33]:*

$$LT_\delta\{\varphi(\mu)\} = \Omega(\mu, w) = \frac{1}{\Gamma(1+\delta)} \int_0^\infty E_\delta\left(-w^\delta \mu^\delta\right) \varphi(\mu) \ (d\mu)^\delta, \tag{2}$$

*where the latter integral converges and $w^\delta \in R^\delta$.*

**Definition 3.** *The inverse formula of the local fractional Laplace transform is given by the following:*

$$LT_\delta^{-1}[\Omega(\mu, w)] = \varphi(\mu) = \frac{1}{(2\pi)^\delta} \int_{\beta - i\infty}^{\beta + i\infty} E_\delta\left(w^\delta \mu^\delta\right) \Omega(\mu, w) \ (dw)^\delta, \tag{3}$$

*where $w^\delta = \beta + i\omega$; fractal imaginary unit $i^\delta$ and $Re(w) = \beta > 0$.*

**Definition 4.** *The Mittage-Leffler function and hyperbolic sine are, respectively, defined by [19,20,32,33]:*

$$E_\delta\left(\mu^\delta\right) = \sum_{k=0}^\infty \frac{\mu^{k\delta}}{\Gamma(1+k\delta)}, \ \mu \in R, \ 0 < \delta \le 1, \tag{4}$$

$$\sinh_\delta\left(\mu^\delta\right) = \frac{E_\delta\left(\mu^\delta\right) - E_\delta\left(-\mu^\delta\right)}{2}, \ \mu \in R, \ 0 < \delta \le 1. \tag{5}$$

*The properties for the local fractional Laplace transform used in the paper are given as follows:*

1. $LT_\delta\left\{\varphi^{(k\delta)}(\mu)\right\} = w^{k\delta} LT_\delta\{\varphi(\mu)\} - w^{(k-1)\delta}\varphi(0) - \cdots - \varphi^{((k-1)\delta)}(0)$.
2. $LT_\delta\left\{\mu^{k\delta}\right\} = \frac{\Gamma(1+k\delta)}{w^{(k+1)\delta}}$.

## 3. Analysis of LFLHPM

The local fractional homotopy perturbation method (LFHPM) has been developed and applied to solve a class of local fractional partial differential equations [31,34]. Based on it, we suggest a new analytical method.

Let us consider the following partial differential equation with local fractional derivative:

$$L_\delta \varphi(\mu, \tau) + R_\delta \varphi(\mu, \tau) = f(\mu, \tau), \ 0 < \delta \le 1, \tag{6}$$

where $L_\delta = \frac{\partial^{k\delta}}{\partial \mu^{k\delta}}$, $R_\delta$ is a linear local fractional operator, and $f(\mu, \tau)$ is the source term.

Applying the local fractional Laplace transform (LFLT) to Equation (6), it gives the following:

$$\Omega(\mu, w) = w^{-\delta}\varphi(0, \tau) + w^{-2\delta}\varphi^{(\delta)}(0, \tau) + \cdots + w^{-(k-1)\delta}\varphi^{((k-1)\delta)}(0, \tau) + w^{-k\delta}LT_\delta\{f(\mu, \tau)\} - w^{-k\delta}LT_\delta\{R_\delta\varphi(\mu, \tau)\}. \tag{7}$$

Using the inversion of LFLT on Equation (7), we have the following:

$$\varphi(\mu, \tau) = G(\mu, \tau) - LT_\delta^{-1}\left[w^{-k\delta}LT_\delta\{R_\delta\varphi(\mu, \tau)\}\right], \tag{8}$$

where $G(\mu, \tau)$ represents the term arising from the source term and the prescribed initial conditions. Now, we apply the LFHPM [31]:

$$\varphi(\mu, \tau) = \sum_{n=0}^{\infty} p^n \varphi_n(\mu, \tau). \tag{9}$$

Using Equation (9) in Equation (8), it yields the following result:

$$\sum_{n=0}^{\infty} p^n \varphi_n(\mu, \tau) = G(\mu, \tau) - p\, \mathrm{LT}_\delta^{-1}\left[w^{-k\delta}\mathrm{LT}_\delta\left\{R_\delta \sum_{n=0}^{\infty} p^n \varphi_n(\mu, \tau)\right\}\right]. \tag{10}$$

On equating the multipliers of same powers of the parameter $p$ of Equation (10), it gives the following:

$$p^0 : \varphi_0(\mu, \tau) = G(\mu, \tau),$$
$$p^1 : \varphi_1(\mu, \tau) = -\mathrm{LT}_\delta^{-1}\left[w^{-k\delta}\mathrm{LT}_\delta\{R_\delta\, \varphi_0(\mu, \tau)\}\right],$$
$$p^2 : \varphi_2(\mu, \tau) = -\mathrm{LT}_\delta^{-1}\left[w^{-k\delta}\mathrm{LT}_\delta\{R_\delta\, \varphi_1(\mu, \tau)\}\right],$$
$$p^3 : \varphi_3(\mu, \tau) = -\mathrm{LT}_\delta^{-1}\left[w^{-k\delta}\mathrm{LT}_\delta\{R_\delta\, \varphi_2(\mu, \tau)\}\right],$$
$$\vdots$$

Proceeding in this same manner, the rest of the components $\varphi_n(\mu, \tau)$ can be completely obtained and the series solution is thus entirely determined. Finally, we approximate the analytical solution $\varphi(\mu, \tau)$ by truncated series as follows:

$$\varphi(\mu, \tau) = \lim_{N \to \infty} \sum_{n=0}^{N} \varphi_n(\mu, \tau). \tag{11}$$

## 4. Application of LFLHPM for Helmholtz Equations

**Example 1.** *Let us consider the local fractional Helmholtz equation with local fractional derivative operator:*

$$\frac{\partial^{2\delta}\varphi(\mu, \tau)}{\partial \mu^{2\delta}} + \frac{\partial^{2\delta}\varphi(\mu, \tau)}{\partial \tau^{2\delta}} + \varphi(\mu, \tau) = \frac{\mu^\delta}{\Gamma(1+\delta)}\frac{\tau^\delta}{\Gamma(1+\delta)}, \quad 0 < \delta \le 1 \tag{12}$$

*with fractal initial boundary conditions*

$$\varphi(0, \tau) = 0, \quad \varphi^{(\delta)}(0, \tau) = \frac{\tau^\delta}{\Gamma(1+\delta)} \quad (0 < \tau \le l). \tag{13}$$

*Applying the LFLT on both sides (12), subject to initial condition (13), we have the following:*

$$\Omega(\mu, w) = w^{-2\delta}\frac{\tau^\delta}{\Gamma(1+\delta)} + w^{-2\delta}\mathrm{LT}_\delta\left\{\frac{\mu^\delta}{\Gamma(1+\delta)}\frac{\tau^\delta}{\Gamma(1+\delta)}\right\} - w^{-2\delta}\mathrm{LT}_\delta\left\{\frac{\partial^{2\delta}\varphi(\mu, \tau)}{\partial \tau^{2\delta}} + \varphi(\mu, \tau)\right\}. \tag{14}$$

*The inversion of LFLT implies that*

$$\varphi(\mu, \tau) = \frac{\mu^\delta}{\Gamma(1+\delta)}\frac{\tau^\delta}{\Gamma(1+\delta)} + \frac{\mu^{3\delta}}{\Gamma(1+3\delta)}\frac{\tau^\delta}{\Gamma(1+\delta)} - \mathrm{LT}_\delta^{-1}\left[w^{-2\delta}\mathrm{LT}_\delta\left\{\frac{\partial^{2\delta}\varphi(\mu, \tau)}{\partial \tau^{2\delta}} + \varphi(\mu, \tau)\right\}\right]. \tag{15}$$

*Now, applying LFHPM, we obtain the following:*

$$\sum_{n=0}^{\infty} p^n \varphi_n(\mu, \tau) = \frac{\mu^\delta}{\Gamma(1+\delta)}\frac{\tau^\delta}{\Gamma(1+\delta)} + \frac{\mu^{3\delta}}{\Gamma(1+3\delta)}\frac{\tau^\delta}{\Gamma(1+\delta)} - p\, \mathrm{LT}_\delta^{-1}\left[w^{-2\delta}\mathrm{LT}_\delta\left\{\frac{\partial^{2\delta}}{\partial \tau^{2\delta}}\left(\sum_{n=0}^{\infty} p^n \varphi_n(\mu, \tau)\right) + \sum_{n=0}^{\infty} p^n \varphi_n(\mu, \tau)\right\}\right]. \tag{16}$$

*Comparing the coefficients of like power of p, we get the following:*

$$p^0 : \varphi_0(\mu, \tau) = \frac{\mu^\delta}{\Gamma(1+\delta)} \frac{\tau^\delta}{\Gamma(1+\delta)} + \frac{\mu^{3\delta}}{\Gamma(1+3\delta)} \frac{\tau^\delta}{\Gamma(1+\delta)},$$

$$p^1 : \varphi_1(\mu, \tau) = -LT_\delta^{-1}\left[w^{-2\delta} LT_\delta\left\{\frac{\partial^{2\delta}}{\partial \tau^{2\delta}} \varphi_0(\mu, \tau) + \varphi_0(\mu, \tau)\right\}\right],$$

$$p^2 : \varphi_2(\mu, \tau) = -LT_\delta^{-1}\left[w^{-2\delta} LT_\delta\left\{\frac{\partial^{2\delta}}{\partial \tau^{2\delta}} \varphi_1(\mu, \tau) + \varphi_1(\mu, \tau)\right\}\right],$$

$$p^3 : \varphi_3(\mu, \tau) = -LT_\delta^{-1}\left[w^{-2\delta} LT_\delta\left\{\frac{\partial^{2\delta}}{\partial \tau^{2\delta}} \varphi_2(\mu, \tau) + \varphi_2(\mu, \tau)\right\}\right],$$

$$\vdots$$

*Hence, we have:*

$$p^0 : \varphi_0(\mu, \tau) = \frac{\mu^\delta}{\Gamma(1+\delta)} \frac{\tau^\delta}{\Gamma(1+\delta)} + \frac{\mu^{3\delta}}{\Gamma(1+3\delta)} \frac{\tau^\delta}{\Gamma(1+\delta)},$$

$$p^1 : \varphi_1(\mu, \tau) = -\frac{\mu^{3\delta}}{\Gamma(1+3\delta)} \frac{\tau^\delta}{\Gamma(1+\delta)} - \frac{\mu^{5\delta}}{\Gamma(1+5\delta)} \frac{\tau^\delta}{\Gamma(1+\delta)},$$

$$p^2 : \varphi_2(\mu, \tau) = \frac{\mu^{5\delta}}{\Gamma(1+5\delta)} \frac{\tau^\delta}{\Gamma(1+\delta)} + \frac{\mu^{7\delta}}{\Gamma(1+7\delta)} \frac{\tau^\delta}{\Gamma(1+\delta)},$$

$$p^3 : \varphi_3(\mu, \tau) = -\frac{\mu^{7\delta}}{\Gamma(1+7\delta)} \frac{\tau^\delta}{\Gamma(1+\delta)} - \frac{\mu^{9\delta}}{\Gamma(1+9\delta)} \frac{\tau^\delta}{\Gamma(1+\delta)},$$

$$\vdots$$

*Therefore, the series solution of Equation (12) is given by the following:*

$$\varphi(\mu, \tau) = \lim_{N \to \infty} \sum_{n=0}^{N} \varphi_n(\mu, \tau) = \frac{\mu^\delta}{\Gamma(1+\delta)} \frac{\tau^\delta}{\Gamma(1+\delta)}. \tag{17}$$

*The result is the same as the one which is obtained by the local fractional variational iteration method* [35].

## 5. Application of LFLHPM for Coupled Helmholtz Equations

**Example 2.** *Let us consider the coupled Helmholtz equations with local fractional derivative operators:*

$$\frac{\partial^{2\delta}\varphi(\mu,\tau)}{\partial\mu^{2\delta}} + \frac{\partial^{2\delta}\psi(\mu,\tau)}{\partial\tau^{2\delta}} - \varphi(\mu,\tau) = 0,$$
$$\frac{\partial^{2\delta}\psi(\mu,\tau)}{\partial\mu^{2\delta}} + \frac{\partial^{2\delta}\varphi(\mu,\tau)}{\partial\tau^{2\delta}} - \psi(\mu,\tau) = 0 \tag{18}$$

*subject to the initial boundary conditions*

$$\varphi(0,\tau) = 0, \ \varphi^{(\delta)}(0,\tau) = E_\delta(\tau^\delta), \ (0 < \tau \leq l)$$
$$\psi(0,\tau) = 0, \ \psi^{(\delta)}(0,\tau) = -E_\delta(\tau^\delta). \ (0 < \tau \leq l). \tag{19}$$

*Applying the LFLT on both sides (18), subject to initial condition (19), we have the following:*

$$\Omega(\mu, w) = w^{-2\delta} E_\delta(\tau^\delta) + w^{-2\delta} LT_\delta\left\{\varphi(\mu,\tau) - \frac{\partial^{2\delta}\psi(\mu,\tau)}{\partial\tau^{2\delta}}\right\},$$
$$\Psi(\mu, w) = -w^{-2\delta} E_\delta(\tau^\delta) + w^{-2\delta} LT_\delta\left\{\psi(\mu,\tau) - \frac{\partial^{2\delta}\varphi(\mu,\tau)}{\partial\tau^{2\delta}}\right\}. \tag{20}$$

*The inversion of LFLT implies that*

$$\varphi(\mu,\tau) = \frac{\mu^\delta}{\Gamma(1+\delta)}E_\delta(\tau^\delta) + LT_\delta^{-1}\left[w^{-2\delta}LT_\delta\left\{\varphi(\mu,\tau) - \frac{\partial^{2\delta}\psi(\mu,\tau)}{\partial\tau^{2\delta}}\right\}\right],$$

$$\psi(\mu,\tau) = -\frac{\mu^\delta}{\Gamma(1+\delta)}E_\delta(\tau^\delta) + LT_\delta^{-1}\left[w^{-2\delta}LT_\delta\left\{\psi(\mu,\tau) - \frac{\partial^{2\delta}\varphi(\mu,\tau)}{\partial\tau^{2\delta}}\right\}\right]$$

(21)

*Now, applying LFHPM, we obtain the following:*

$$\varphi(\mu,\tau) = \sum_{n=0}^{\infty} p^n \varphi_n(\mu,\tau),$$

$$\psi(\mu,\tau) = \sum_{n=0}^{\infty} p^n \psi_n(\mu,\tau).$$

(22)

*Using Equation (22) in Equation (21), it yields the following result:*

$$\sum_{n=0}^{\infty} p^n \varphi_n(\mu,\tau) = \frac{\mu^\delta}{\Gamma(1+\delta)}E_\delta(\tau^\delta) + p\,LT_\delta^{-1}\left[w^{-2\delta}LT_\delta\left\{\sum_{n=0}^{\infty} p^n \varphi_n(\mu,\tau) - \frac{\partial^{2\delta}}{\partial\tau^{2\delta}}\left(\sum_{n=0}^{\infty} p^n \psi_n(\mu,\tau)\right)\right\}\right],$$

$$\sum_{n=0}^{\infty} p^n \psi_n(\mu,\tau) = -\frac{\mu^\delta}{\Gamma(1+\delta)}E_\delta(\tau^\delta) + p\,LT_\delta^{-1}\left[w^{-2\delta}LT_\delta\left\{\sum_{n=0}^{\infty} p^n \psi_n(\mu,\tau) - \frac{\partial^{2\delta}}{\partial\tau^{2\delta}}\left(\sum_{n=0}^{\infty} p^n \varphi_n(\mu,\tau)\right)\right\}\right].$$

*Comparing the coefficients of like power of p, we get the following:*

$$p^0 : \begin{cases} \varphi_0(\mu,\tau) = \frac{\mu^\delta}{\Gamma(1+\delta)}E_\delta(\tau^\delta), \\ \psi_0(\mu,\tau) = -\frac{\mu^\delta}{\Gamma(1+\delta)}E_\delta(\tau^\delta), \end{cases}$$

$$p^1 : \begin{cases} \varphi_1(\mu,\tau) = LT_\delta^{-1}\left[w^{-2\delta}LT_\delta\left\{\varphi_0(\mu,\tau) - \frac{\partial^{2\delta}}{\partial\tau^{2\delta}}\psi_0(\mu,\tau)\right\}\right], \\ \psi_1(\mu,\tau) = LT_\delta^{-1}\left[w^{-2\delta}LT_\delta\left\{\psi_0(\mu,\tau) - \frac{\partial^{2\delta}}{\partial\tau^{2\delta}}\varphi_0(\mu,\tau)\right\}\right], \end{cases}$$

$$p^2 : \begin{cases} \varphi_2(\mu,\tau) = LT_\delta^{-1}\left[w^{-2\delta}LT_\delta\left\{\varphi_1(\mu,\tau) - \frac{\partial^{2\delta}}{\partial\tau^{2\delta}}\psi_1(\mu,\tau)\right\}\right], \\ \psi_2(\mu,\tau) = LT_\delta^{-1}\left[w^{-2\delta}LT_\delta\left\{\psi_1(\mu,\tau) - \frac{\partial^{2\delta}}{\partial\tau^{2\delta}}\varphi_1(\mu,\tau)\right\}\right], \end{cases}$$

$$\vdots$$

*Hence, we have:*

$$p^0 : \begin{cases} \varphi_0(\mu,\tau) = \frac{\mu^\delta}{\Gamma(1+\delta)}E_\delta(\tau^\delta), \\ \psi_0(\mu,\tau) = -\frac{\mu^\delta}{\Gamma(1+\delta)}E_\delta(\tau^\delta), \end{cases}$$

$$p^1 : \begin{cases} \varphi_1(\mu,\tau) = \frac{2\mu^{3\delta}}{\Gamma(1+3\delta)}E_\delta(\tau^\delta), \\ \psi_1(\mu,\tau) = -\frac{2\mu^{3\delta}}{\Gamma(1+3\delta)}E_\delta(\tau^\delta), \end{cases}$$

$$p^2 : \begin{cases} \varphi_2(\mu,\tau) = \frac{4\mu^{5\delta}}{\Gamma(1+5\delta)}E_\delta(\tau^\delta), \\ \psi_2(\mu,\tau) = -\frac{4\mu^{5\delta}}{\Gamma(1+5\delta)}E_\delta(\tau^\delta), \end{cases}$$

$$p^2 : \begin{cases} \varphi_2(\mu,\tau) = \frac{8\mu^{7\delta}}{\Gamma(1+7\delta)}E_\delta(\tau^\delta), \\ \psi_2(\mu,\tau) = -\frac{8\mu^{7\delta}}{\Gamma(1+7\delta)}E_\delta(\tau^\delta), \end{cases}$$

$$\vdots$$

*and so on, and the other components can be found in a similar manner. Therefore, the series solutions can be written in the following form:*

$$
\begin{aligned}
\varphi(\mu,\tau) &= \lim_{N\to\infty} \sum_{n=0}^{N} \varphi_n(\mu,\tau) \\
&= E_\delta(\tau^\delta)\left[\frac{\mu^\delta}{\Gamma(1+\delta)} + \frac{2\mu^{3\delta}}{\Gamma(1+\delta)} + \frac{4\mu^{5\delta}}{\Gamma(1+\delta)} + \cdots\right] \\
&= E_\delta(\tau^\delta)\frac{sinh_\delta(\sqrt{2}\mu^\delta)}{\sqrt{2}}. \\
(\mu,\tau) &= \lim_{N\to\infty} \sum_{n=0}^{N} \psi_n(\mu,\tau) \\
&= -E_\delta(\tau^\delta)\left[\frac{\mu^\delta}{\Gamma(1+\delta)} + \frac{2\mu^{3\delta}}{\Gamma(1+\delta)} + \frac{4\mu^{5\delta}}{\Gamma(1+\delta)} + \cdots\right] \\
&= -E_\delta(\tau^\delta)\frac{sinh_\delta(\sqrt{2}\mu^\delta)}{\sqrt{2}}.
\end{aligned}
$$

*The result is the same as the one which is obtained by the local fractional variational iteration transform method* [36].

## 6. Conclusions

In this work, the LFLHPM was successfully applied to finding the approximate solution of Helmholtz and coupled Helmholtz equations involving LFDOs. A comparison was made to show that the method has a small computational size in comparison with the computational size required in other numerical methods, such as the local fractional variational iteration method and the local fractional variational iteration transform method. The method is very powerful and efficient in finding analytical as well as numerical solutions for wide classes of linear and nonlinear local fractional PDEs.

**Author Contributions:** H.K.J. wrote some sections of the manuscript; D.B. prepared some other sections of the paper and analyzed. All authors have read and approved the final manuscript.

**Funding:** This research received no external funding.

**Conflicts of Interest:** The authors declare no conflict of interest

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
