# Peer review of "A Modification Fractional Homotopy Perturbation Method for Solving Helmholtz and Coupled Helmholtz Equations on Cantor Sets"

_fractalfract, doi:10.3390/fractalfract3020030_

Reviewer 1 Report

This paper deals with the application on local fractional Laplace homotopy peturbation method on Helmholtz equations. The paper is organized as follows. In Section 2, the authors recall some mathematical results. They describe their method in Section 3 and apply it in Section 4.

The paper is easy to read, and well organized. However, I have the following comments that the authors must take into account.

-Introduction: What is the point of writing 10 acronyms (LFDM, LFRDTM...) without explaining them and giving the relative advantages of each method. This is an introduction, not a catalog. Moreover, the reader may not be familiar with these acronyms.

-Section 2: the variable w is not defined. Maybe it should read LT_\delta{\phi(u)}(w)? Is the local fractional Laplace transform always defined? What about its inverse? Please be more rigorous. For known results, you should add references.

-Section 3: what about \phi(0,t) in equation (3.3)? Why do you have a p that appear in equation (3.5) (G(\mu, \tau)-pLT_\delta^{-1}[...])? I really do not understand it.

-Section 4: Please give more details on how you obtained the term \frac{\mu^{3\delta} \tau^\delta}{\Gamma(1+3\delta) \Gamma(1+\delta)} in (4.4).

-Conclusion. You write that your method is very powerful and efficient for wide classes of linear and non-linear local fractional PDEs. Actually, you simply applied it on only one class of PDE, without doing any simulation results and comparison with other existing method. What about the computational time? The accuracy compared to other methods? This needs to be improved

-General comment: please check typos and English ("and to used it" in the intro, "calculus are introduction" in the intro.

All in all I cannot recommend acceptance.

Author Response

Dear Sir

Kindly, the reply in the attached file

Reviewer 2 Report

The paper presents a new analytical technique for solving Helmholtz equation. The method is applied to two examples, one partial differential equation and one coupled partial differential equation. 

The mathematical definitions that base the method and are important to write the solutions are stated. Some foundation works are cited. Nonetheless, some references involving methods for solving Helmholtz equation are missing, namely:

M.S SAMUEL,.; A. THOMAS. On fractional Helmholtz equations. Fractional Calculus & Applied Analysis. V. 13 (3) (2010) 295-308. 

E. GOLDFAIN,.; Fractional dynamics, Cantorian space–time and the gauge hierarchy problem. Chaos, Solitons and Fractals 22 (2004) 513–520.

VALENTIM JR, C. A.; BANNWART, F. DE C.; DAVID, S. A. Fractional calculus applied to linear thermoacoustics : A generalization of Rott’s model. 17th Brazillian Congress of Thermal Sciences and Engineering (ENCIT). Anais... SP - Brazil: ABCM, 2018.

AI-MIN YANG, ZENG-SHUN CHEN, H. M. SRIVASTAVA, AND XIAO-JUN YANG, Application of the Local Fractional Series Expansion Method and the Variational Iteration Method to the Helmholtz Equation Involving Local Fractional Derivative Operators. Abstract and Applied Analysis Volume 2013, Article ID 259125, 1-6.

The method proposed seems interesting enough to pave an alternative for finding relatively simple and elegant solutions to a large group of differential equations. It will be interesting to see it being applied to models governed by Helmholtz equations from different areas.

There aren’t any problems regarding concepts and mathematics as far as I could verify. 

The main ideas of the paper are well presented. The sentences are objective and correctly linked. The vocabulary is adequate and technical. However, there are several grammatical issues that should be corrected. Some examples:

Line 28 “In recent years, a many of approximate…”

Line 30 “…LFHPM [31]. our aim…”

Line 32 “… and to used it to solve…”

Line 35/36 “… local fractional calculus are introduction.”

Problems like these are consistent throughout the text. 

I believe that the paper should be accepted with minor revisions after the following improvements: English language, inclusion of aforementioned references and typos correction.

Author Response

Dear Sir 

Kindly, the reply in attached file

Reviewer 3 Report

The paper needs major revisions. Please see the attached file.

Author Response

Dear Sir 

Kindly, the reply in attached file

Round  2

Reviewer 1 Report

Thank you for the corrections.

Reviewer 3 Report

The authors have improved their paper, and I recommend it for publication.